# Features of patients who developed hepatocellular carcinoma after direct-acting antiviral treatment for hepatitis C Virus

Seiichi Mawatari[1]*, Kotaro Kumagai[1], Kohei Oda[1], Kazuaki Tabu[1], Sho Ijuin[1], Kunio Fujisaki[2], Shuzo Tashima[1,2], Yukiko Inada[3], Hirofumi Uto[1,3], Akiko Saisyoji[1,4], Yasunari Hiramine[4], Masafumi Hashiguchi[1,5], Tsutomu Tamai[1,5], Takeshi Hori[5], Ohki Taniyama[1], Ai Toyodome[1], Haruka Sakae[1], Takeshi Kure[1,6], Kazuhiro Sakurai[7], Akihiro Moriuchi[1,7], Shuji Kanmura[1], Akio Ido[1]

1 Department of Human and Environmental Sciences, Digestive and Lifestyle Diseases, Kagoshima University Graduate School of Medical and Dental Sciences, Kagoshima, Japan, 2 Department of Hepatology, Kirishima Medical Center, Hayato-cho, Kirishima, Kagoshima, Japan, 3 Center for Digestive and Liver Diseases, Miyazaki Medical Center Hospital, Miyazaki, Japan, 4 Department of Internal Medicine, Kagoshima Kouseiren Hospital, Kagoshima, Japan, 5 Department of Gastroenterology and Hepatology, Kagoshima City Hospital, Kagoshima, Japan, 6 Department of Gastroenterology, Kagoshima Medical Association Hospital, Kagoshima, Japan, 7 Department of Gastroenterology, National Hospital Organization Kagoshima Medical Center, Kagoshima, Japan

* mawatari@m2.kufm.kagoshima-u.ac.jp

**Data Availability Statement:** All relevant data are within the manuscript and its Supporting Information files (S1 File).

## Abstract

### Background

The features of hepatitis C virus patients with a sustained virologic response (SVR) who developed hepatocellular carcinoma (HCC) after direct-acting antiviral (DAA) therapy are unclear.

### Methods

The study population included 1494 DAA-SVR patients without a history of HCC. The cumulative carcinogenesis rate after the end of treatment (EOT) and factors related to HCC were analyzed.

### Results

Sixty (4.0%) patients developed HCC during a median observation period of 47.6 months. At four years, the cumulative carcinogenesis rate was 4.7%. A Cox proportional hazards analysis showed that age ≥73 years (hazard ratio [HR]: 2.148), male sex (HR: 3.060), hyaluronic acid (HA) ≥75 ng/mL (HR: 3.996), alpha-fetoprotein at EOT (EOT-AFP) ≥5.3 ng/mL (HR: 4.773), and albumin at EOT (EOT-Alb) <3.9 g/dL (HR: 2.305) were associated with HCC development. Especially, EOT-AFP ≥5.3 ng/mL was associated with HCC development after 3 years from EOT (HR: 6.237). Among patients who developed HCC, AFP did not increase in patients with EOT-AFP <5.3 ng/mL at the onset of HCC. Of these 5 factors, EOT-AFP ≥5.3 ng/mL was scored as 2 points; the others were scored as 1 point. The 4-

**Funding:** This work was supported in part by a grant-in-aid from the Ministry of Health, Labour and Welfare of Japan (grant number: 18K15821). The funders had no role in study design, data collection and analysis, decision to publish, or preparation of the manuscript.

**Competing interests:** I have read the journal's policy and the authors of this manuscript have the following competing interests: AI received honoraria for lectures from Bristol-Myers Squibb Co., Ltd., MSD Co., Ltd., Gilead Sciences Co., Ltd., and Abbvie Inc., and received research funding from Eisai Co., Ltd., Bristol-Myers Squibb Co., Ltd, MSD Co., Ltd., and Abbvie Inc. The other authors declare no conflicts of interest in association with the present study. This does not alter our adherence to PLOS ONE policies on sharing data and materials.

year cumulative carcinogenesis rate for patients with total scores of 0–2, 3–4, and 5–6 points were 0.6%, 11.9%, and 27.1%, respectively (p<0.001).

## Conclusions

EOT-AFP $\geq$5.3 ng/mL is useful for predicting HCC development after an SVR. However, AFP does not increase in patients with EOT-AFP <5.3 ng/mL at the onset of HCC. The combination of EOT-AFP, age, sex, HA, and EOT-Alb is important for predicting carcinogenesis.

## Introduction

Chronic hepatitis C virus (HCV) infection affects 71 million people worldwide and approximately 399,000 people die each year from hepatitis C-related liver disease [1, 2]. In Japan, approximately 30,000 people died of hepatocellular carcinoma (HCC) in 2016. In 2007, the major etiology was persistent HCV infection, which accounted for 65% of all HCC deaths [3]. In 2011, it is estimated that the population of people with HCV infection in Japan was 0.98–1.6 million [3].

One of the goals of therapy is to cure HCV infection in order to prevent the complications of HCV-related liver diseases, including hepatic necroinflammation, fibrosis, cirrhosis, decompensated cirrhosis, HCC, and death [1].

In recent years, direct-acting antivirals (DAAs) have been approved, and IFN-free therapy with DAAs has achieved very high sustained virologic response (SVR) rates of $\geq$95% [2]. DAA treatment has been frequently administered to elderly or cirrhotic patients, and achieved high SVR rates [4–7]. However, the characteristics of patients who develop HCC during long-term observation are unclear. In the present study, we aimed to clarify the features of patients who developed HCC after a DAA-SVR.

## Materials and methods

### Study population

This was a prospective observational study conducted at 21 facilities belonging to the Kagoshima Liver Study Group in Japan. Study population enrollment is shown in Fig 1. In brief, a total of 1521 patients with chronic HCV infection and no history of HCC therapy were treated with DAAs, and achieved an SVR between October 2014 and December 2019, and was observed until July 2021. We excluded patients who had HBV co-infection, those who were confirmed to have tumors in the liver or other organs, and those who developed HCC during DAA treatment. Liver tumors included hypovascular tumors, such as dysplastic nodules or well-differentiated HCC diagnosed by contrast-enhanced (CE) computed tomography (CT) or magnetic resonance imaging (MRI) before therapy. Ultimately, 1494 patients were analyzed in this study. Written informed consent was obtained from the enrolled patients. The study protocol conformed to the ethical guidelines of the Declaration of Helsinki and was approved by the Kagoshima University Hospital Clinical Research Ethics Committee and the research ethics committee of each participating facility (approval numbers: 150138, 170199, 190297).

The HCV RNA concentration was measured by TaqMan PCR, which has a lower quantitation limit of 1.2 log IU/mL. The Fib-4 index, a surrogate marker of liver fibrosis, was calculated based on the methods of previous studies [8]. Liver cirrhosis was comprehensively judged by

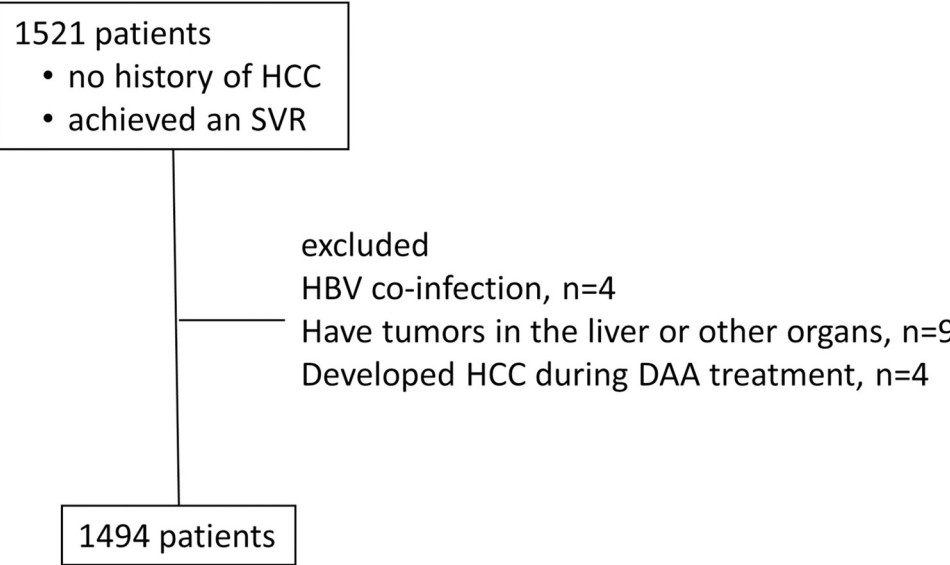

**Fig 1. Flow chart of enrollment of the study population.** HCC, hepatocellular carcinoma; DAA, direct acting antiviral.

hepatologists at each institution according to the platelet count, imaging, fibrosis markers, transient elastography, or varices formation.

## Treatment protocol

Patients were treated with daclatasvir (DCV) and asunaprevir (ASV) for 24 weeks, sofosbuvir (SOF) and ledipasvir (LDV) for 12 weeks, ombitasvir (OBV) and paritaprevir (PTV) and ritonavir (r) for 12 weeks, SOF and ribavirin (RBV) for 12 weeks, elbasvir (EBR) and grazoprevir (GZR) for 12 weeks, DCV and ASV and beclabuvir (BCV) for 12 weeks, and glecaprevir (GLE) and pibrentasvir (PIB) for 8 or 12 weeks. All patients were treated according to Japanese guidelines for chronic HCV infection [9]. The initiation of the observation period was defined as the end of treatment (EOT) with DAAs.

## Surveillance of HCC

The Japanese guidelines state that cirrhotic patients have an extremely high risk of developing HCC and should be monitored every 3–4 months, and that non-cirrhotic patients have a high risk of developing HCC and be monitored every 6 months by ultrasonography (US), CT or MRI [10]. HCC was diagnosed when typical vascular findings were observed by contrast-enhanced CT or MRI, which showed hyper-enhancement in the arterial phase and a washout pattern in the portal, or delayed phases. Abdominal US, CE-CT or CE-MRI were performed at 3–6-month intervals.

## Statistical analyses

Statistical analyses were performed using the IBM Statistical Package for Social Sciences (SPSS) software program (version 26 IBM SPSS Statistics, Armonk, NY, USA). Categorical data were compared using the chi-squared test and Fisher's exact test, as appropriate. Continuous variables were analyzed using the Mann-Whitney U test. The Kaplan–Meier method and log rank test were used to analyze the cumulative rates of HCC development. Correlation coefficients were obtained using Spearman's rank correlation coefficient. P values of <0.05 were

considered to indicate statistical significance. Factors associated with HCC development were determined using a Cox proportional hazards analysis with forward selection using p<0.10 as a cutoff for inclusion in the model. For the categorical data, we determined the cut-off values at which the optimal sensitivity and specificity were achieved using receiver operating characteristic (ROC) curves.

## Results

### Baseline characteristics

Table 1 shows the baseline characteristics of the patients. The average age was 67.1 years, 603 of patients were men (40.4%), 357 patients had cirrhosis (23.9%), 644 patients had a Fib-4 index of ≥3.25, which suggested advanced fibrosis (23.9%) [8], and 59 patients had a history of DAA treatment for HCV (3.9%). The median observation period was 47.6 months.

### Comparison of the baseline characteristics between the patients who developed HCC and non-HCC patients

Sixty (4.0%) of 1494 patients developed HCC (Table 1). The cumulative rates of HCC development were 1.1% at 1 year, 1.8% at 2 years, 3.5% at 3 years, 4.7% at 4 years, and 5.6% at 5 years

**Table 1. Patient characteristics.**

| Characteristics | Total n = 1494 | HCC development (-) n = 1434 | HCC development (+) n = 60 | P value |
|---|---|---|---|---|
| Age, years (range) | 67.1±10.8 (26–90) | 66.9±10.8 (26–90) | 71.5±9.2 (48–87) | 0.001 |
| Male, n (%) | 603 (40.4) | 566 (39.5) | 37 (61.7) | <0.001 |
| Liver cirrhosis, n (%) | 357 (23.9) | 324 (22.6) | 33 (55.0) | <0.001 |
| Body Mass Index, kg/m2 (n = 1179) | 22.8±3.5 | 22.9±3.5 | 22.0±2.5 | 0.059 |
| Prior DAA therapy, none/ experience, n (%) | 1435/ 59 | 1377/ 57 | 58/ 2 | 0.574 |
| Diabetes Mellitus, n (%) | 249 (16.7) | 240 (16.7) | 9 (15.0) | 0.444 |
| Genotype 1/ 2/ 1+2 | 1196/ 297/ 1 | 1140/ 293/ 1 | 56/ 4/ 0 | 0.032 |
| DCV+ASV/ SOF/LDV/ OBV/PTV/r/ SOF+RBV/ | 362/ 480/ 108/ 198 | 340/ 455/ 106/ 194 | 22/ 25/ 2/ 4 | 0.015 |
| GZR+EBR/ DCV/ASV/BCV/ GLE/PIB, n | 103/ 12/ 231 | 98/ 12/ 229 | 5/ 0/ 2 | |
| HCV-RNA, logIU/mL | 6.0±0.9 | 6.0±0.9 | 5.9±0.7 | 0.171 |
| Platelet counts, ×$10^4$/μL | 15.8±5.8 | 15.9±5.8 | 12.3±6.3 | <0.001 |
| Total bilirubin, mg/dL (n = 1491) | 0.8±0.4 | 0.8±0.4 | 1.0±0.6 | 0.002 |
| AST, U/L | 49±34 | 49±34 | 58±37 | <0.001 |
| ALT, U/L | 49±45 | 49±46 | 50±35 | 0.276 |
| GGT, U/L | 45±53 | 45±52 | 61±73 | 0.003 |
| Alb, g/dL (n = 1457) | 4.1±0.4 | 4.1±0.4 | 3.8±0.5 | <0.001 |
| Fib-4 index | 3.82±2.98 | 3.72±2.88 | 6.31±4.16 | <0.001 |
| Fib-4 index >3.25, n (%) | 644 (43.1) | 599 (41.8) | 45 (75.0) | <0.001 |
| Hyaluronic acid, (n = 1415) | 166.2±271.5 | 162.4±274.0 | 252.2±190.2 | <0.001 |
| AFP, ng/mL (n = 1475) | 9.5±31.5 | 9.3±31.8 | 15.8±25.1 | <0.001 |
| DCP, mAU/mL (n = 1059) | 22.3±23.1 | 22.3±23.2 | 26.5±20.8 | 0.462 |
| EOT-ALT, non WNL | 170 (11.4) | 159 (11.1) | 11 (18.3) | 0.070 |
| EOT-Alb, g/dL (n = 1413) | 4.1±0.4 | 4.1±0.4 | 3.8±0.4 | <0.001 |
| EOT-AFP, ng/mL (n = 1385) | 4.6±7.6 | 4.4±6.8 | 9.3±17.6 | <0.001 |

Data are shown as the mean ± standard deviation, HCC, hepatocellular carcinoma; DAA, direct-acting antivirals; DCV, daclatasvir; ASV, asunaprevir; SOF, sofosbuvir; LDV, ledipasvir; OBV, ombitasvir; PTV, paritaprevir; r, ritonavir; RBV, ribavirin; GZR, grazoprevir; EBR, elbasvir; BCV, beclabuvir; GLE, glecaprevir; PIB, pibrentasvir; HCV, hepatitis C virus; AST, aspartate transaminase; ALT, alanine transaminase; GGT, γ-glutamyltransferase; Alb, albumin; AFP, α-fetoprotein; DCP, des-γ-carboxy prothrombin; EOT, end of treatment; WNL, within normal limit.

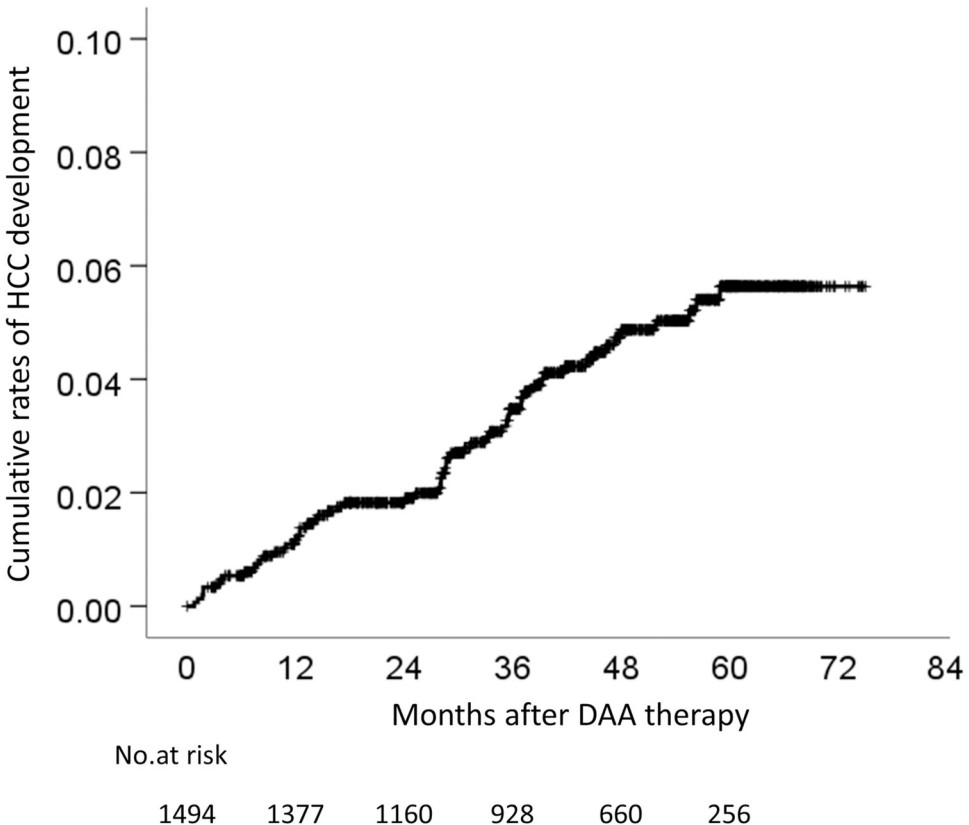

**Fig 2. Cumulative rates of HCC development in patients with DAA-SVR.** HCC, hepatocellular carcinoma; DAA, direct-acting antiviral; SVR, sustained virologic response.

(Fig 2). The comparison of the baseline characteristics between patients who developed HCC and non-HCC revealed that the patients who developed HCC were older, were more frequently male, and had a higher incidence of cirrhosis in comparison to the non-HCC patients (Table 1). Regarding the blood test results of the HCC patients, the total bilirubin, aspartate aminotransferase (AST), gamma-glutamyl transpeptidase (GGT), hyaluronic acid (HA), Fib-4 index, and alpha-fetoprotein (AFP) values were significantly higher, and the albumin level, and platelet count were significantly lower (Table 1). In addition, the AFP level at the EOT (EOT-AFP) was significantly as higher, and albumin at the EOT (EOT-Alb) were significantly lower (Table 1). However, the des-γ-carboxy prothrombin (DCP) level did not differ to a statistically significant extent.

## Setting the cut-off values for continuous variables associated with hepatocarcinogenesis

For the categorical data, we determined the cut-off values at which the optimal sensitivity and specificity were achieved using ROC curves (Table 2). The EOT-AFP had the higher area under ROC and positive likelihood ratio (Table 2).

## Factors associated with the development of HCC in DAA-treated patients who achieved an SVR

The Cox proportional hazard analysis showed that the following factors were associated with the development of HCC: age ≥73 years (hazard ratio [HR]: 2.148), male sex (HR: 3.060), HA

**Table 2. Setting cutoff values for continuous variables associated with hepatocarcinogenesis.**

| | Cut off value | Sensitivity (%) | Specificity (%) | PPV (%) | NPV (%) | Likelihood ratio | odds ratio | AUC |
|---|---|---|---|---|---|---|---|---|
| Age | 73 years | 53.3 | 69.9 | 6.9 | 97.3 | 1.774 | 2.660 | 0.624 |
| AST | 47 U/L | 60.0 | 65.6 | 6.8 | 97.5 | 1.742 | 2.854 | 0.631 |
| ALT | 36 U/L | 60.0 | 51.9 | 5.0 | 96.9 | 1.247 | 1.617 | 0.541 |
| GGT | 31 U/L | 63.3 | 54.4 | 5.5 | 97.3 | 1.389 | 2.060 | 0.614 |
| Total bilirubin | 0.8 mg/dL | 58.3 | 60.3 | 5.8 | 97.1 | 1.467 | 2.121 | 0.620 |
| Platelet | $11.3 \times 10^4$ /µL | 53.3 | 78.1 | 9.2 | 97.6 | 2.428 | 4.060 | 0.683 |
| Alb | 3.9 g/dL | 63.3 | 74.2 | 9.5 | 97.9 | 2.451 | 4.957 | 0.697 |
| Hyaluronic acid | 75 ng/mL | 88.1 | 47.8 | 6.8 | 98.9 | 1.688 | 6.799 | 0.714 |
| Fib-4 index | 4.03 | 68.3 | 69.4 | 8.4 | 98.0 | 2.193 | 4.578 | 0.732 |
| AFP | 7.1 ng/mL | 56.7 | 73.9 | 8.4 | 97.6 | 2.173 | 3.707 | 0.696 |
| EOT-Alb | 3.9 g/dL | 55.9 | 77.9 | 9.9 | 97.6 | 2.533 | 4.478 | 0.693 |
| EOT-AFP | 5.3 ng/mL | 65.0 | 78.3 | 11.9 | 98.0 | 2.980 | 6.657 | 0.732 |
| 5 risk factors† | 2 | 81.0 | 76.0 | 13.7 | 98.8 | 3.373 | 13.513 | 0.838 |

† The sum of scores when EOT-AFP ≥5.3 ng/mL was scored as 2 points; the others (i.e., age ≥73 years, male sex, hyaluronic acid ≥75 ng/mL, and EOT-Alb <3.9 g/dL) were scored as 1 point.

PPV, positive predictive value; NPV, negative predictive value; AUC, area under the receiver operator characteristic curve; AST, aspartate transaminase; ALT, alanine transaminase; GGT, γ-glutamyltransferase; Alb, albumin; AFP, α-fetoprotein; EOT, end of treatment.

≥75 ng/mL (HR: 3.996), EOT-AFP ≥5.3 ng/mL (HR: 4.773), and EOT-Alb <3.9 g/dL (HR: 2.305) (Table 3). In the analysis of 564 patients with Fib-4 index ≥3.25 which suggested advanced fibrosis [8], 45 patients developed HCC, and male sex (HR: 2.986), EOT-AFP ≥5.3 ng/mL (HR: 8.352), and EOT-Alb <3.9 g/dL (HR: 2.803) were associated with the

**Table 3. Factors associated with the development of hepatocellular carcinoma (HCC) in all patients and severe fibrosis.**

| Category | Cut off | Univariate | Multivariate (All patients, n = 1284) | | | Multivariate (Fib-4 index >3.25, n = 564) | | |
|---|---|---|---|---|---|---|---|---|
| | | P value | P value | Hazard Ratio | 95%CI | P value | Hazard Ratio | 95%CI |
| Age | ≥73 years | 0.001 | 0.005 | 2.148 | 1.264–3.651 | 0.068 | 1.774 | 0.958–3.288 |
| Sex | Male | <0.001 | <0.001 | 3.060 | 1.788–5.237 | <0.001 | 2.986 | 1.588–5.615 |
| Etiology | Cirrhosis present | <0.001 | | | | 0.072 | 1.885 | 0.946–3.758 |
| Prior DAA therapy | DAA experience | 0.574 | | | | | | |
| Platelet counts | <$11.3 \times 10^4$ /µL | <0.001 | | | | | | |
| Total bilirubin | ≥0.8 mg/dL | 0.006 | | | | | | |
| AST | ≥47 U/L | <0.001 | | | | | | |
| ALT | ≥36 U/L | 0.078 | | | | | | |
| GGT | ≥31 U/L | 0.006 | | | | | | |
| Alb | <3.9 g/dL | <0.001 | | | | | | |
| Hyaluronic acid | ≥75 ng/mL | <0.001 | 0.002 | 3.996 | 1.668–9.571 | | | |
| Fib-4 index | ≥4.03 | <0.001 | | | | | | |
| AFP | ≥7.1 ng/mL | <0.001 | | | | 0.064 | 0.457 | 0.199–1.047 |
| EOT-ALT | Not WNL | 0.070 | | | | | | |
| EOT-Alb | <3.9 g/dL | <0.001 | 0.003 | 2.305 | 1.336–3.977 | 0.003 | 2.803 | 1.425–5.517 |
| EOT-AFP | ≥5.3 ng/mL | <0.001 | <0.001 | 4.773 | 2.718–8.383 | <0.001 | 8.352 | 3.314–21.047 |

95%CI, 95% confidence interval; DAA, direct-acting antivirals; AST, aspartate transaminase; ALT, alanine transaminase; GGT, γ-glutamyltransferase; Alb, albumin; AFP, α-fetoprotein; EOT, end of treatment; WNL, within normal limit.

**Table 4. Factors associated with the development of hepatocellular carcinoma (HCC) in over 6-month or 1-year follow up model.**

| Category | Cut off | Univariate | Multivariate (after 6 month, n = 1245) | | | Multivariate (after 1 year, n = 1195) | | |
|---|---|---|---|---|---|---|---|---|
| | | *P* value | *P* value | Hazard Ratio | 95%CI | *P* value | Hazard Ratio | 95%CI |
| Age | ≥73 years | 0.001 | 0.045 | 1.782 | 1.014–3.132 | 0.057 | 1.816 | 0.983–3.355 |
| Sex | Male | <0.001 | <0.001 | 2.835 | 1.605–5.007 | <0.001 | 3.088 | 1.652–5.772 |
| Etiology | Cirrhosis present | <0.001 | | | | | | |
| Prior DAA therapy | DAA experience | 0.574 | | | | | | |
| Platelet counts | <11.3×10⁴ /μL | <0.001 | | | | | | |
| Total bilirubin | ≥0.8 mg/dL | 0.006 | | | | | | |
| AST | ≥47 U/L | <0.001 | | | | | | |
| ALT | ≥36 U/L | 0.078 | | | | | | |
| GGT | ≥31 U/L | 0.006 | | | | | | |
| Alb | <3.9 g/dL | <0.001 | 0.019 | 2.084 | 1.127–3.856 | | | |
| Hyaluronic acid | ≥75 ng/mL | <0.001 | 0.007 | 3.481 | 1.411–8.583 | 0.009 | 3.300 | 1.343–8.105 |
| Fib-4 index | ≥4.03 | <0.001 | | | | | | |
| AFP | ≥7.1 ng/mL | <0.001 | | | | | | |
| EOT-ALT | Not WNL | 0.070 | | | | | | |
| EOT-Alb | <3.9 g/dL | <0.001 | | | | 0.007 | 2.289 | 1.218–4.303 |
| EOT-AFP | ≥5.3 ng/mL | <0.001 | <0.001 | 4.257 | 2.352–7.704 | <0.001 | 3.288 | 1.754–6.163 |

95%CI, 95% confidence interval; DAA, direct-acting antivirals; AST, aspartate transaminase; ALT, alanine transaminase; GGT, γ-glutamyltransferase; Alb, albumin; AFP, α-fetoprotein; EOT, end of treatment; WNL, within normal limit.

development of HCC (Table 3). In the analysis of 1245 patients who did not develop HCC within 6 months and who were observed for more than 6 months, 51 patients developed HCC, and age ≥73 years (HR: 1.782), male sex (HR: 2.835), HA ≥75 ng/mL (HR: 3.481), EOT-AFP ≥5.3 ng/mL (HR: 4.257), and Alb <3.9 g/dL (HR: 2.084) were associated with the development of HCC (Table 4). In the analysis of 1195 patients who did not develop HCC within 1 year and who were observed for more than 1 year, 43 patients developed HCC, and male sex (HR: 3.088), HA ≥75 ng/mL (HR: 3.300), EOT-AFP ≥5.3 ng/mL (HR: 3.288), and EOT-Alb <3.9 g/dL (HR: 2.289) were associated with the development of HCC (Table 4).

Similarly, in the analysis of 1031 patients who did not develop HCC within 2 years and who were observed for more than 2 years, 34 patients developed HCC, and male sex (HR: 2.326), HA ≥75 ng/mL (HR: 4.085), EOT-AFP ≥5.3 ng/mL (HR: 4.272), and EOT-Alb <3.9 g/dL (HR: 2.352) were associated with HCC development (Table 5). In addition, in the analysis of 814 patients who did not develop HCC within 3 years and who were observed for more than 3 years, 16 patients developed HCC, and cirrhosis (HR: 5.775) and EOT-AFP ≥5.3 ng/mL (HR: 6.237) were associated with the development of HCC (Table 5).

## Comparison of the cumulative rates of HCC development

The cumulative rates of HCC development according to each cutoff value were compared. The cumulative rates of HCC development of patients who were ≥73 and <73 years of age were 2.2% and 0.5%, respectively, at one year, 3.9% and 0.7% at two years, 6.1% and 2.1% at three years, 7.6% and 3.3% at four years (Fig 3A). The cumulative rates of HCC development in male and female patients were 1.7% and 0.7%, respectively, at one year, 3.2% and 0.9% at two years, 6.2% and 1.7% at three years, and 7.2% and 3.2% at four years (Fig 3B). The cumulative rates of HCC development in patients with cirrhosis and non-cirrhosis were 3.2% and 0.5%, respectively, at one year, 4.4% and 1.0% at two years, 6.8% and 2.5% at three years, and 10.6%

**Table 5. Factors associated with the development of hepatocellular carcinoma (HCC) in over 2-year or 3-year follow up model.**

| Category | Cut off | Univariate | Multivariate (after 2 years, n = 1031) | | | Multivariate (after 3 years, n = 814) | | |
|---|---|---|---|---|---|---|---|---|
| | | P value | P value | Hazard Ratio | 95%CI | P value | Hazard Ratio | 95%CI |
| Age | ≥73 years | 0.001 | | | | | | |
| Sex | Male | <0.001 | 0.017 | 2.326 | 1.163–4.652 | | | |
| Etiology | Cirrhosis present | <0.001 | | | | 0.003 | 5.775 | 1.792–18.613 |
| Prior DAA therapy | DAA experience | 0.574 | | | | | | |
| Platelet counts | <11.3×10$^4$/μL | <0.001 | | | | | | |
| Total bilirubin | ≥0.8 mg/dL | 0.006 | | | | | | |
| AST | ≥47 U/L | <0.001 | | | | | | |
| ALT | ≥36 U/L | 0.078 | | | | | | |
| GGT | ≥31 U/L | 0.006 | | | | | | |
| Alb | <3.9 g/dL | <0.001 | | | | | | |
| Hyaluronic acid | ≥75 ng/mL | <0.001 | 0.011 | 4.085 | 1.389–12.015 | | | |
| Fib-4 index | ≥4.03 | <0.001 | | | | | | |
| AFP | ≥7.1 ng/mL | <0.001 | | | | | | |
| EOT-ALT | Not WNL | 0.070 | | | | | | |
| EOT-Alb | <3.9 g/dL | <0.001 | 0.019 | 2.352 | 1.151–4.807 | | | |
| EOT-AFP | ≥5.3 ng/mL | <0.001 | <0.001 | 4.272 | 2.061–8.858 | 0.002 | 6.237 | 1.933–20.123 |

95%CI, 95% confidence interval; DAA, direct-acting antivirals; AST, aspartate transaminase; ALT, alanine transaminase; GGT, γ-glutamyltransferase; Alb, albumin; AFP, α-fetoprotein; EOT, end of treatment; WNL, within normal limit.

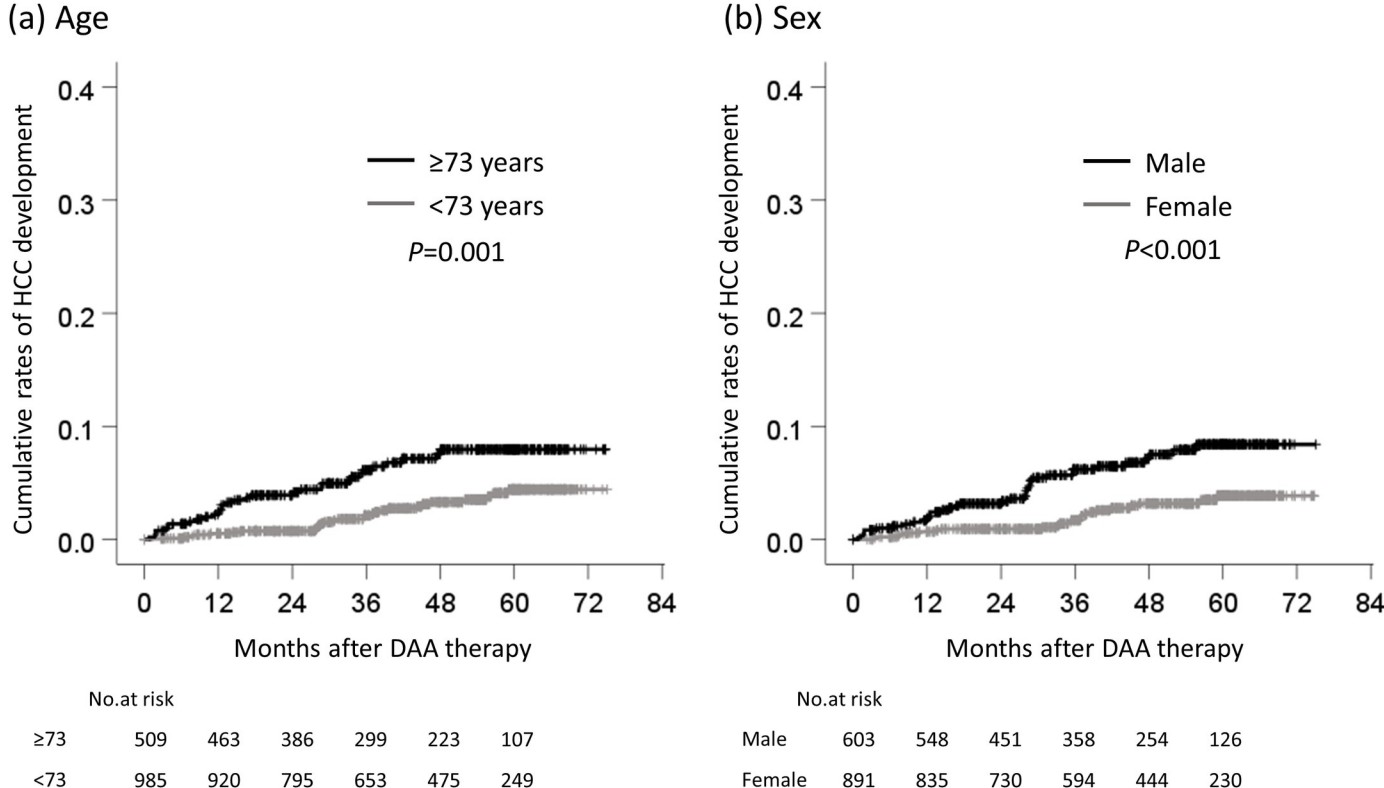

**Fig 3. The cumulative rates of HCC development by age and sex.** The comparison of the cumulative rates of HCC development (a) between ≥73 years of age and <73 years of age, (b) between male and female. HCC, hepatocellular carcinoma; DAA, direct-acting antiviral.

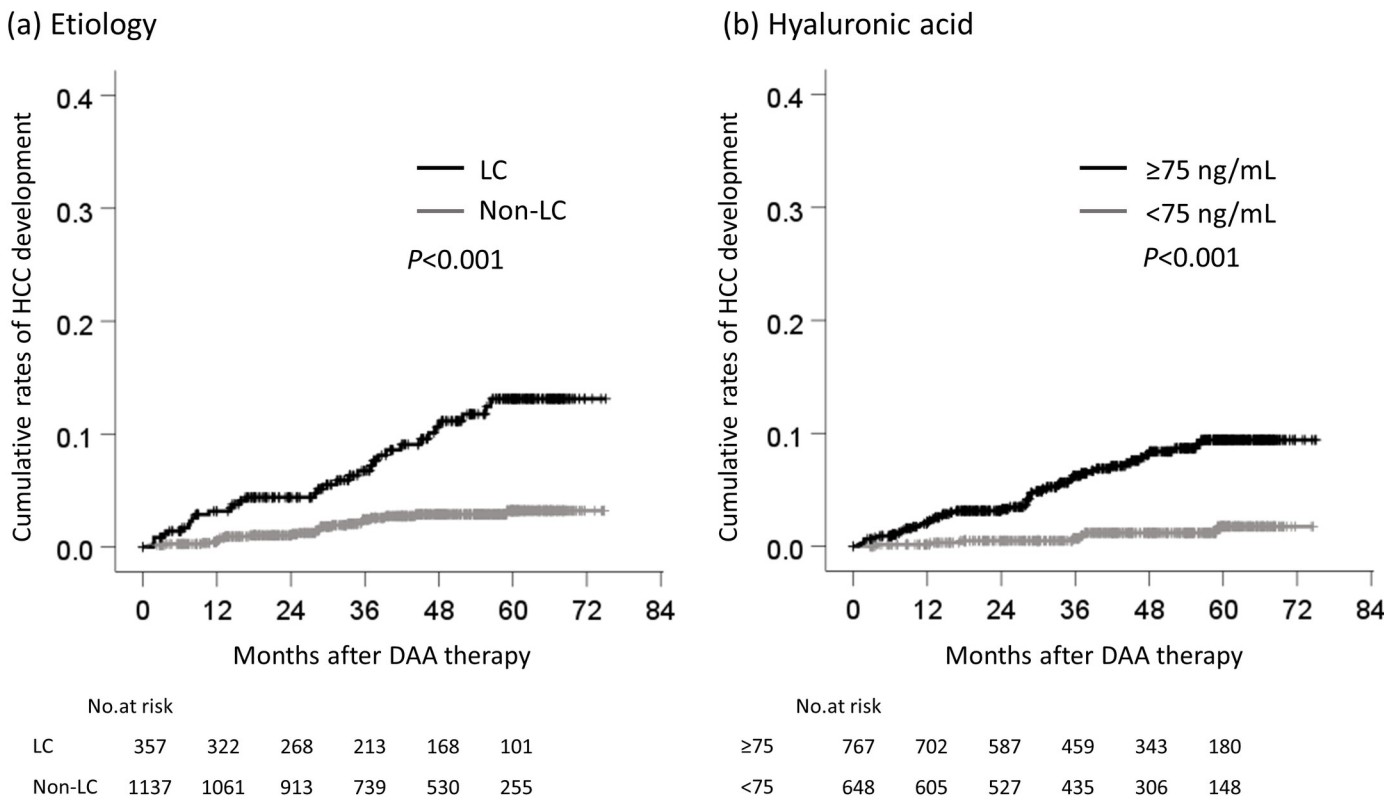

**(a) Etiology**

**(b) Hyaluronic acid**

No.at risk

| | | | | | | |
|---|---|---|---|---|---|---|
| LC | 357 | 322 | 268 | 213 | 168 | 101 |
| Non-LC | 1137 | 1061 | 913 | 739 | 530 | 255 |

No.at risk

| | | | | | | |
|---|---|---|---|---|---|---|
| ≥75 | 767 | 702 | 587 | 459 | 343 | 180 |
| <75 | 648 | 605 | 527 | 435 | 306 | 148 |

**Fig 4. The cumulative rates of HCC development by etiology and hyaluronic acid.** (a) between cirrhosis and non-cirrhosis, (b) between HA ≥75 ng/mL and <75 ng/mL. HCC, hepatocellular carcinoma; DAA, direct-acting antiviral; LC, liver cirrhosis; HA, hyaluronic acid.

and 2.9% at four years (Fig 4A). The cumulative rates of HCC development in patients with HA ≥75 and <75 ng/mL were 2.0% and 0.2%, respectively, at one year, 3.1% and 0.5% at two years, 6.3% and 0.7% at three years, and 8.1% and 1.2% at four years (Fig 4B). The cumulative rates of HCC development in patients with EOT-Alb <3.9 and ≥3.9 g/dL were 3.1% and 0.5%, respectively, at one year, 4.7% and 1.0% at two years, 9.3% and 1.8% at three years, and 11.9% and 2.7% at four years (Fig 5A). Finally, the cumulative rates of HCC development in patients with EOT-AFP ≥5.3 and <5.3 ng/mL were 4.3% and 0.2%, respectively, at one year, 5.2% and 0.9% at two years, 9.4% and 1.9% at three years, and 12.8% and 2.4% at four years (Fig 5B).

### Relationship between changes in AFP and the development of HCC

Since EOT-AFP was strongly related to the development of HCC, we investigated the relationship between the transition of AFP and HCC development. The patients were classified into four groups according to their AFP before treatment and at the EOT, and the cumulative rates of HCC development were examined in each group. The four groups according to the AFP levels were as follows, (a) <5.3 ng/mL before treatment and <5.3 ng/mL at the EOT, (b) ≥5.3 ng/mL, <5.3 ng/mL, (c) <5.3 ng/mL, ≥ 5.3 ng/mL, and (d) ≥5.3 ng/mL, ≥5.3 ng/mL. The cumulative rates of HCC development at 1, 2, 3, and 4 years were as follows: (a) 0.2%, 1.0%, 1.7%, 2.1%, (b) 0%, 0.5%, 2.7%, 3.5%, (c) 3.8%, 8.0%, 18.8%, 30.6%, (d) 4.3%, 5.0%, 8.6%, 11.8%, respectively (Fig 6). There were statistically significant differences in the rates of HCC development between groups (a) and (c), (a) and (d), (b) and (c), and (b) and (d) (p<0.001); however, there were no significant differences between groups (a) and (b), or (c) and (d) (Fig 6).

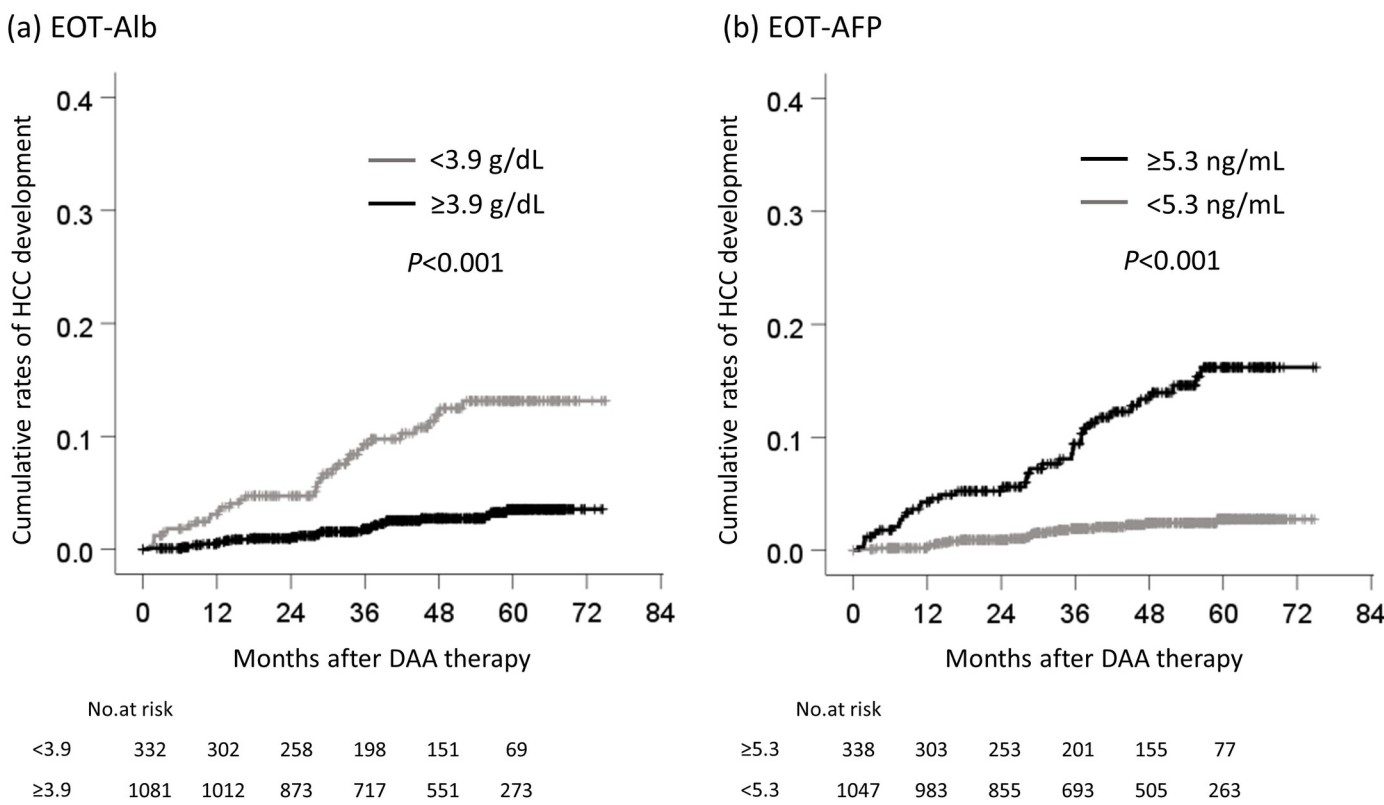

**Fig 5. The cumulative rates of HCC development by albumin and alpha-fetoprotein at the end of treatment.** (a) between EOT-Alb ≥3.9 g/dL and <3.9g/dL, (b) between EOT-AFP ≥5.3 ng/mL and <5.3 ng/mL. HCC, hepatocellular carcinoma; DAA, direct-acting antiviral; EOT, end of treatment; Alb, albumin; AFP, alpha-fetoprotein.

### Collinearity between hyaluronic acid and cirrhosis or advanced fibrosis

HA was related to the development of HCC. We therefore investigated the relationship between HA and cirrhosis, or advanced fibrosis. Patients with liver cirrhosis or severe fibrosis had significantly higher HA values than non-cirrhotic patients or patients with non-severe fibrosis (p<0.001, respectively, Fig 7A and 7B). Patients of >73 years of age had significantly higher HA values than who were <73 years of age (p<0.001, Fig 7C). However, there was no sex difference in the HA values (p = 0.649) (Fig 7D). HA showed a significant positive correlation with the Fib-4 index (correlation coefficient [rs] = 0.615, p<0.001) (Fig 8).

### Development of HCC stratified by the score combining EOT-AFP, age, sex, hyaluronic acid, and EOT-Alb

A scoring system was developed based on the results of the multivariate analysis that included the EOT-AFP, age, sex, HA, and EOT-Alb. Of these five factors, EOT-AFP ≥5.3 ng/mL was associated with the development of HCC from 3 years after the EOT; thus, EOT-AFP ≥5.3 ng/mL was scored as 2 points, and others, such as age ≥73 years, male sex, HA ≥75 ng/mL, and EOT-Alb <3.9 g/dL were scored as 1 point. When the cutoff value of the new score was 2, the area under ROC, positive likelihood ratio, and odds ratio all increased (Table 2). The patients were divided into 3 groups based on the sum of these scores, as follows: 0–2 points (low-risk), 3–4 points (moderate-risk), and 5–6 points (high-risk). The cumulative carcinogenesis rate of each group was examined. The cumulative rates of HCC development in the low-, moderate-,

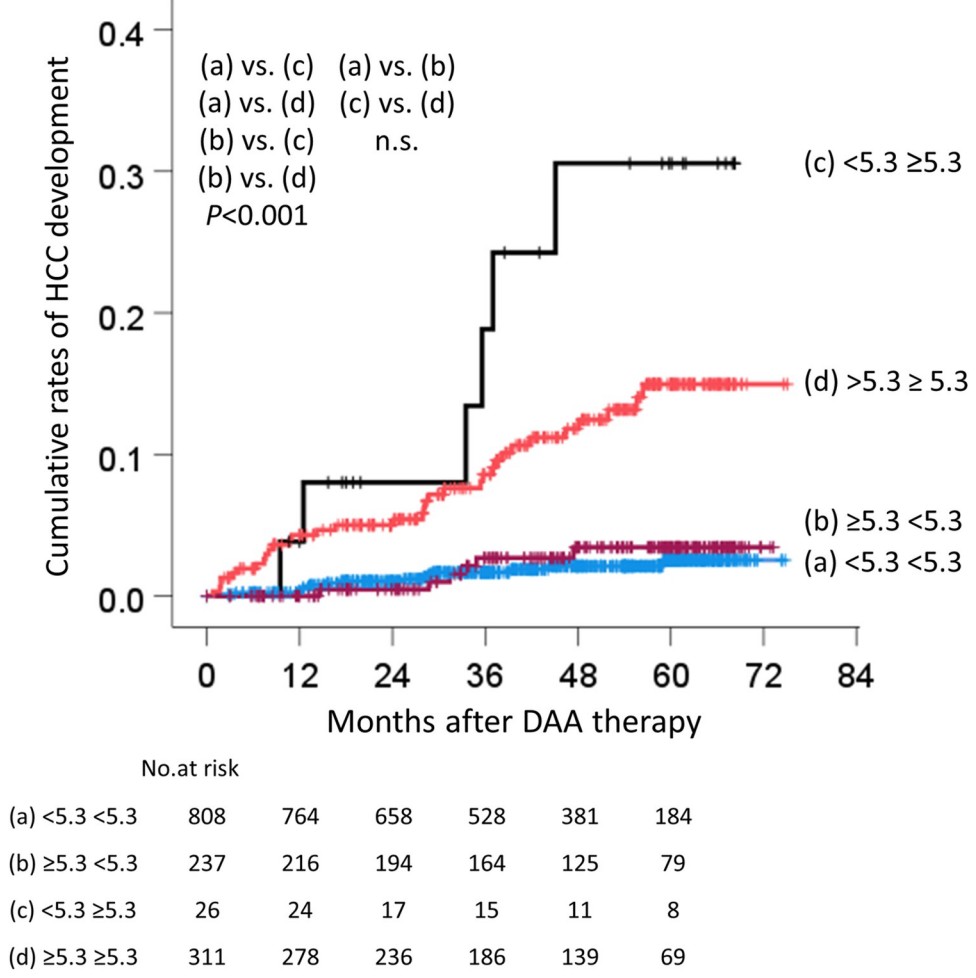

**Fig 6. The cumulative rates of HCC development for each change of AFP.** AFP was classified into 4 groups before treatment and at the EOT: (a) <5.3 ng/mL before treatment, and <5.3 ng/mL at the EOT, (b) ≥5.3 ng/mL, <5.3 ng/mL, (c) <5.3 ng/mL, ≥5.3 ng/mL, and (d) ≥5.3 ng/mL, ≥5.3 ng/mL. HCC, hepatocellular carcinoma; DAA, direct-acting antiviral; EOT, end of treatment; AFP, α-fetoprotein.

and high-risk groups were 0%, 1.5%, and 12.4% respectively, at one year, 0.3%, 3.3%, and 15.0% at two years, 0.4%, 7.8% and 25.3% at three years, and 0.6%, 11.9% and 27.2% at four years (Fig 9). There were statistically significant differences between the groups in the rate of HCC development (p<0.001).

## Alpha-fetoprotein and des-γ-carboxy prothrombin levels at the onset of HCC

AFP and des-γ-carboxy prothrombin (DCP) are tumor markers for HCC. In 60 patients who developed HCC, we confirmed the AFP and DCP levels at the onset of HCC. The details are summarized in Table 6. The median tumor size and median number of tumors was 1.8 cm and 1 nodule, respectively, 93.1% of patients developed HCC with vascularity. The comparison of the characteristics between patients with EOT-AFP ≥5.3 ng/mL and <5.3 ng/mL revealed that there was no significant difference in tumor size, number, or vascularity, the patients with EOT-AFP ≥5.3 ng/mL were more frequently male, and higher AFP and DCP levels. Especially,

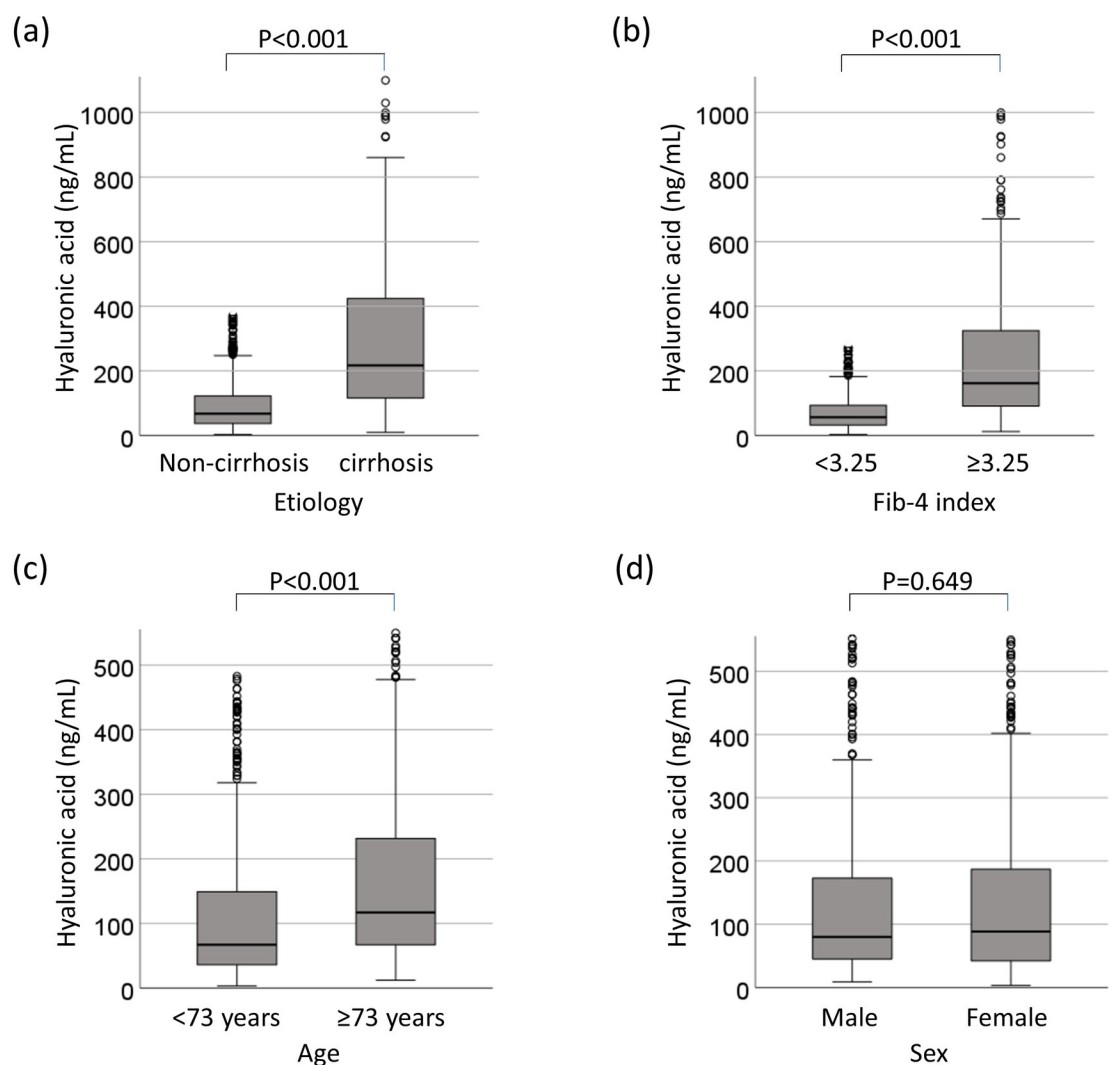

**Fig 7. Collinearity between hyaluronic acid and cirrhosis or advanced fibrosis.** (a) Comparison of HA in patients with and without cirrhosis. (b) Comparison of HA in patients with Fib4-index values of ≥3.25 and <3.25, (c) Comparison of HA in patients of ≥73 years of age and <73 years of age, (d) Comparison of HA in males and females. HA, hyaluronic acid.

twenty-three patients (38.3%) had AFP levels above the upper limit (10 ng/mL), and 27 patients (45.8%) had DCP levels above the upper limit (40 mAU/mL). Twenty-two of 39 patients (56.4%) with EOT-AFP ≥5.3 ng/mL had levels above the upper limit at the onset of HCC. However, only 1 of 21 patients (4.8%) with EOT-AFP <5.3 ng/mL had a level above the upper limit (p<0.001). Similarly, AFP increased more than 30% at the onset of HCC in 19 of 39 patients (48.7%) with EOT-AFP ≥5.3 ng/mL, but only increased in two of 21 patients (9.5%) with EOT-AFP <5.3 ng/mL (p<0.001). On the other hand, among patients with EOT-AFP <5.3 ng/mL, the percentage of patients with DCP levels above the upper limit at the onset of HCC was similar to that of patients with EOT-AFP ≥5.3 ng/mL (p = 0.477). In addition, 22 patients (37.3%) had AFP and DCP levels within the normal limits at the onset of HCC. Among patients with EOT-AFP <5.3 ng/mL, a significantly higher percentage of patients had normal limits of AFP and DCP in comparison to those with EOT-AFP ≥5.3 ng/ mL at the onset of HCC (p = 0.020).

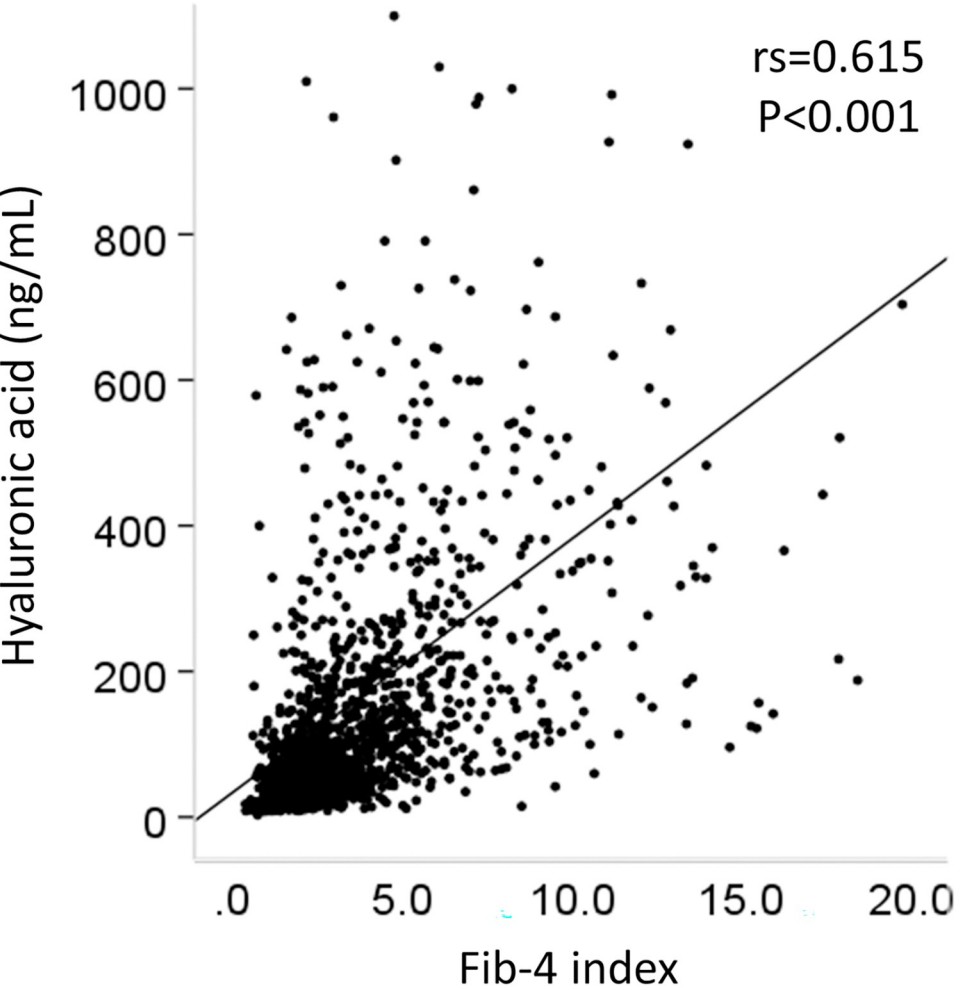

**Fig 8. Correlation between HA and the Fib-4 index.** rs, correlation coefficient.

## Discussion

In the present study, we revealed that age, sex, HA, EOT-AFP, and EOT-Alb were useful markers for predicting the development of HCC in patients who achieved an SVR with DAA treatment for HCV. By combining these factors, the risk of developing HCC could be significantly stratified (Fig 9). The median observation period was 47.6 months, with approximately 60% patients observed for more than 3 years. The advantage of our study is that the predictors were derived from a long observation period.

Since the EOT-AFP was associated with the development of HCC from 3 years after the EOT (Table 5), we investigated the relationship between the transition of AFP and the development of HCC. First, we focused on the transition of AFP before treatment and at the EOT. We revealed that there was no significant difference in the rate of HCC development between patients in whom AFP decreased to <5.3 ng/mL and those in whom AFP remained <5.3 ng/mL at the EOT (Fig 6). Next, we focused on the transition of AFP in patients who developed HCC. In the present study, 56.4% of patients who developed HCC with EOT-AFP ≥5.3 ng/mL had AFP levels above the upper limits at the onset of HCC (Table 6). However, among patients with EOT-AFP <5.3 ng/mL, the percentage of patients with levels above the upper limit or

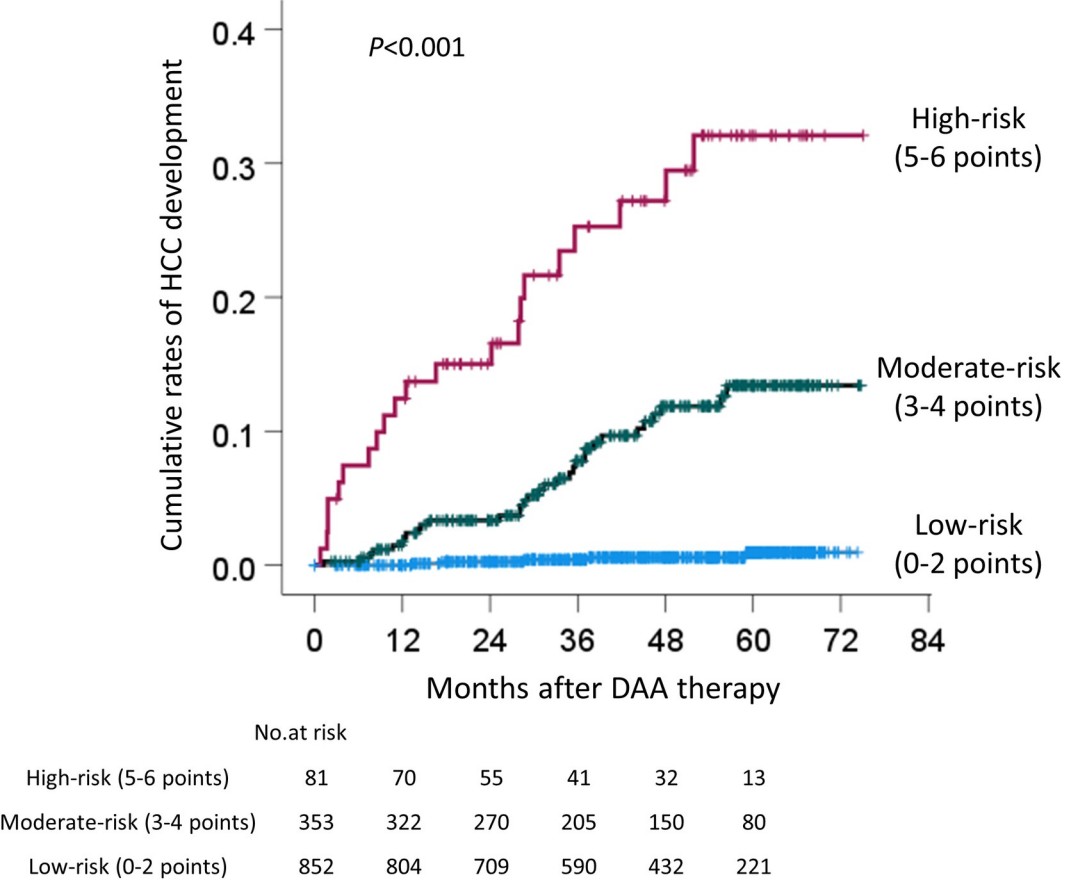

**Fig 9. Development of HCC stratified by the score combining EOT-AFP, age, sex, hyaluronic acid, and EOT-Alb.**
EOT-AFP ≥5.3 ng/mL was scored as 2 points; the others (i.e., age ≥73 years, male sex, HA ≥75 ng/mL, and EOT-Alb <3.9 g/dL) were scored as 1 point. The patients were divided into the following 3 groups according to the sum of these scores: of 0–2 points (low-risk), 3–4 points (moderate-risk), and 5–6 points (high-risk). HCC, hepatocellular carcinoma; EOT, end of treatment; AFP, α-fetoprotein; HA, hyaluronic acid; Alb, albumin.

with a >30% increase in AFP at the onset of HCC were lower in comparison to patients with EOT-AFP ≥5.3 ng/mL (Table 6). In other words, in patients with EOT-AFP <5.3 ng/mL, AFP did not increase, even if they developed HCC; thus the measurement of AFP after an SVR is unlikely to predict the development of HCC. In fact, 35% of patients who developed HCC had EOT-AFP values of <5.3 ng/mL. There were no significant differences between the two groups with respect to tumor size, number and vascularity.In these cases, by combining HA, EOT-Alb, sex, and age, it is possible to identify patients with a high risk of HCC development (Fig 9). To the best of our knowledge, no other studies have focused on the relationship between the AFP transition and carcinogenesis. In addition, in 40.9% of patients with EOT-AFP <5.3 ng/mL, DCP is above the upper limit at the onset of HCC (Table 6). The comparison of the baseline characteristics of patients who developed HCC and patients who did not develop HCC revealed was no significant difference in the DCP level; thus, it is considered useful to measure DCP in the clinical course. However, it should be noted that in 37.3% of patients who developed HCC, the AFP and DCP values were both within the normal limits (Table 6). It seems difficult to predict carcinogenesis in patients who have achieved a DAA-SVR from tumor markers alone.

**Table 6. Comparison of the data at the onset of hepatocellular carcinoma (HCC).**

| At the onset of HCC | ALL | EOT-AFP <5.3ng/mL | EOT-AFP ≥5.3ng/mL | *P* value |
|---|---|---|---|---|
| | n = 60 | n = 21 | n = 39 | |
| Age, years | 73.9±8.8 | 75.8±8.8 | 72.9±8.8 | 0.201 |
| Male, n (%) | 37 (61.7) | 17 (77.3) | 20 (51.3) | 0.022 |
| Tumor size, cm (median) | 2.2±1.8 (1.8) | 2.6±2.8 (1.8) | 2.0±0.8 (1.8) | 0.738 |
| Number of nodules, n (median) | 1.3±0.7 (1) | 1.4±0.9 (1) | 1.3±0.6 (1) | 0.467 |
| Presence of vascularity, n (%) [a] (n = 58) | 54 (93.1) | 19 (95.0) | 35 (92.1) | 0.572 |
| Platelet counts, ×10⁴/μL | 13.3±6.1 | 15.6±6.9 | 12.1±5.4 | 0.045 |
| Total bilirubin, mg/dL | 1.0±0.6 | 1.0±0.9 | 1.0±0.5 | 0.269 |
| AST, U/L | 29.5±10.6 | 27.5±9.7 | 30.6±11.1 | 0.212 |
| ALT, U/L | 20.8±9.2 | 18.4±7.1 | 22.1±10.0 | 0.209 |
| Albumin, g/dL | 4.0±0.5 | 3.9±0.6 | 4.0±0.5 | 0.735 |
| AFP, ng/mL | 114.7±453.0 | 4.34±8.0 | 174.0±555.1 | <0.001 |
| DCP, mAU/mL (n = 59) | 276.1±924.3 | 126.6±362.0 | 358.6±1117.4 | 0.027 |
| AFP ≥10ng/mL, n (%) | 23 (38.3) | 1 (4.8) | 22 (56.4) | <0.001 |
| AFP ≥30% increase, n (%) | 21 (35.0) | 2 (9.5) | 19 (48.7) | 0.002 |
| DCP ≥40 mAU/mL, n (%) (n = 59) | 27 (45.8) | 9 (40.9) | 18 (47.4) | 0.477 |
| AFP and DCP within normal limit, n (%) (n = 59) [b] | 22 (37.3) | 12 (57.1) | 10 (26.3) | 0.020 |

[a] One patient had hypovascular and hypervascular nodules. In one patient, the presence of vascularity was not evaluated due to renal failure.

[b] The normal limit was defined as AFP < 10 ng/mL, and DCP <40 mAU/mL

Data are shown as the mean ± standard deviation, EOT, end of treatment; AFP, α-fetoprotein; DCP, des-γ-carboxy prothrombin; EOT, end of treatment.

AFP is a glycoprotein with a molecular weight of 67 kDa, which was found in human fetal serum by Bergstrand in 1956 [11]. AFP is not only a tumor marker of HCC, but also a marker of the induction of hepatic progenitor cells [12–14]. In addition, the serum AFP level reflects the function of the liver stem or progenitor cells in patients with acute liver failure [15]. as well as inflammation and fibrosis in patients with chronic hepatitis B [16]. In the present study, the AFP level before DAA therapy was not associated with the development of HCC. Additionally, there was no significant difference in the rate of HCC development between decrease to AFP <5.3 ng/mL and remain AFP <5.3 ng/mL at the EOT (Fig 6). The AFP levels before treatment may reflect liver regeneration, inflammation, and fibrosis, as well as microscopic HCC. DAA treatment may reduce inflammation and may reveal tumor-derived AFP. Alternatively, the proliferative activity of hepatocytes may cause hepatocarcinogenesis [17].

HA is an acidic mucopolysaccharide that is widely distributed in the connective tissue of the body, and is produced in the liver by activated hepatic stellate cells. During chronic liver inflammation, there is continuous hepatic stellate cell activation and therefore increased HA synthesis [18]. HA is related to liver fibrosis. It is a part of many non-invasive algorithms that are used to asses liver fibrosis (e.g., ELF score [19] and Hepascore [20]). In this study, the HA levels of patients with liver cirrhosis were higher than those of non-cirrhotic patients and showed a positive correlation with the Fib-4 index (Fig 8). A previous report showed that the initiation of liver cancer requires the inhibition of p53 by CD44 (a receptor of HA)-enhanced growth factor signaling [21]. HA can be a predictive marker for the development of HCC after DAA-SVR.

Several reports have shown that the elimination of the HCV by interferon (IFN)-based therapy suppresses the development of HCC; the cumulative rate of carcinogenesis after an SVR is reported to be 2.3–8.8% at 5 years and 3.1–11.1% at 10 years, depending on patient characteristics [22]. As is observed with IFN-based therapy, the elimination of HCV induced by DAA

treatment reduces the risk of HCC and mortality [23–28]. On the other hand, some reports have indicated that DAA treatment promotes the development of HCC [29–31]. In the present study, the cumulative rates or HCC development were 4.7% at 4 years, with a median observation period of 47.6 months (Fig 2), and 10.6% at 4 years in cirrhotic patients (Fig 4A). In addition, risk factors for hepatocarcinogenesis after an SVR in patients who receive IFN-based therapy include older age, advanced liver fibrosis, male sex, the AFP level after treatment, glucose metabolism disorders, lipid metabolism disorders and alcohol intake, and other factors [17, 22, 32]. Similar risk factors are expected with DAA treatment [33–35]. Wisteria floribunda agglutinin-positive Mac-2 binding protein (M2BPGi), a marker of liver fibrosis, predicts the early occurrence of HCC after an SVR in DAA-treated patients [36, 37]. In the present study, EOT-AFP and HA had a high predictive ability for HCC development. Even if EOT-AFP was <5.3 ng/mL, it is possible to identify patients with a high risk of HCC development with the combination of HA, EOT-Alb, sex, and age (Fig 9). Actually, new score system according to the sum of these risk factors showed the higher area under ROC, positive likelihood ratio, and odds ratio (Table 2). When the patients were divided into the low-risk (0–2 points), moderate-risk (3–4 points), and high-risk (5–6 points) groups, in comparison to the low-risk group, the 4-year cumulative carcinogenesis rate was 45.3 times higher in the high-risk group and 19.8 times higher in the moderate-risk group (Fig 9). Only 5 patients in the low-risk group developed HCC, there was only one patient with 0 points, while 4 patients who had 2 points. The factors were as follows: age and HA, male and HA, age and male, and EOT-AFP (n = 1 each). In the future, long-term follow-up is necessary for patients with these 5 risk factors, and the usefulness of these cut-off values will need to be validated in other cohorts.

This study performed HCC screening based on the Japanese guidelines. The American Association for the Study of Liver Diseases (AASLD) [38], The European Association for the Study of the Liver (EASL) [39], and The Asian Pacific Association for the Study of the Liver (APASL) [40] guidelines recommend ultrasound screening every 6 months. In particular, the AASLD guideline shows that the risk of HCC is significantly lower in patients without cirrhosis in comparison to those with cirrhosis, and surveillance is not recommended for these patients [38]. However, in this study, out of 60 patients who developed HCC, 27 patients (45%) were non-cirrhotic and 15 (25%) had a Fib-4 index of <3.25 (Table 1); thus, HCC surveillance after DAA-SVR should be considered important, even for non-cirrhotic patients. The EASL Clinical Practice Guidelines show that non-cirrhotic F3 patients, regardless of etiology, may be considered for surveillance based on an individual risk assessment, and in patients who were treated viral chronic hepatitis, there was no evidence for a timing or stiffness threshold to stop surveillance in patients who were included in surveillance programs [39]. In this study, a few patients in the low and moderate-risk groups developed HCC. In the future, it will be necessary to consider the cases in which surveillance can be stopped after DAA-SVR.

An early high incidence of HCC was observed in DAA-treated patients with hypovascular tumors, such as dysplastic nodules [41–43]. We also reported that hypovascular tumors developed into hepatocellular carcinoma at a high rate despite the elimination of HCV by DAA treatment [44]. In the present study, we examined the occurrence of *de novo* HCC, and therefore excluded patients with hypovascular tumors diagnosed by CT or MRI before treatment. However, the doubling time of HCC is reported to be 100 days, and it theoretically takes approximately 9 years for a 10-μm HCC to become a 10-mm lesion that can be detected by diagnostic imaging [45]. In other words, the involvement of AFP in hepatocarcinogenesis several years later may indicate the presence of microscopic HCC.

HIV seropositivity accelerates the progression of fibrosis related to chronic hepatitis C [46]. A nationwide survey in Japan revealed that nearly one-fifth of HIV-positive patients are co-infected with HCV. The determination of HCV genotypes revealed that genotype 3 or 4,

which is rarely seen in HCV mono-infected patients in Japan, was found in a substantial fraction of HIV-infected patients [47]. In this cohort, there were no patients who received the therapy for HIV, and with HCV genotype 3 or 4. We therefore considered that there were no HIV-HCV co-infected patients in this cohort.

HCC occurs as a result of hepatic inflammation and/or changes in the tumor microenvironment. HCV clearance can suppress fibrosis and reduces the incidence of HCC [48]. It is assumed that microRNAs (e.g., miRNA-122) and inflammatory cytokines (e.g., transforming growth factor [TGF]-beta, vascular endothelial growth factor [VEGF], and interleukin-6) are involved in this inflammation, and microRNAs are involved in HCV clearance and HCC development [49–51]. HCV clearance limits fibrosis and reduces the incidence of HCC by switching TGF-beta signaling from fibro-carcinogenesis to tumor suppression [52]. Otherwise, the administration of DAAs induces an early increase in serum VEGF and a change in the inflammatory pattern, which coincides with HCV clearance [53]. In this study, we have not been able to investigate the onset of HCC with microRNAs and inflammatory cytokines; however, we consider this to be an issue for future study.

The present study was associated with several limitations. First, various factors (alcohol intake, obesity, metabolic syndrome, and aspirin use, etc.) were not examined after treatment. Second, we could not diagnose fibrosis by histological or non-invasive methods, such as transient elastography; thus, cirrhosis was likely to have been underdiagnosed. Therefore, we adopted fibrosis markers, such as the Fib-4 index and HA before DAA treatments. Third, other fibrosis markers, such as type IV collagen or M2BPGi, could not be measured.

In conclusion, EOT-AFP $\geq$5.3 ng/mL is a useful marker for predicting the development of HCC after an SVR; however, the AFP level did not increase in patients with EOT-AFP <5.3 ng/mL at the onset of HCC. The combination of EOT-AFP, age, sex, HA, and EOT-Alb is important for predicting carcinogenesis.

## Supporting information

**S1 File. Analysis data set.** All patients' data sets were included in the following file. (XLSX)

## Acknowledgments

The present study was carried out in the following 21 facilities (Kagoshima Liver Study Group): Kagoshima University Hospital, Kirishima Medical Center, Miyazaki Medical Center Hospital, Kagoshima Kouseiren Hospital, Kagoshima City Hospital, Saiseikai Sendai Hospital, Kohshinkai Ogura Hospital, Ikeda Hospital, Izumi General Medical Center, Oshima Hospital, Ibusuki Medical Center, Kagoshima medical center, Hirono Clinic, Kagoshima Teishin Hospital, Satsunan Hospital, Nagaki Clinic, Dr. NAKANISHI's office, Southern Region Hospital, Tanegashima Medical Center, Fujimoto General Hospital, and Nakayama Clinic. We thank the following investigators: Yasushi Imamura (Kagoshima Kouseiren Hospital), Dai Imanaka (Ikeda Hospital), Toshihiro Fujita (Oshima Hospital), Kengo Tsuneyoshi (Oshima Hospital), Akihiko Oshige (Ibusuki Medical Center), Shuichi Hirono (Hirono Clinic), Masahito Nagaki (Nagaki Clinic), Chihiro Nakanishi (Dr. NAKANISHI's office), and Toshihiro Nakayama (Nakayama Clinic). We also thank Ms. Hiromi Eguchi, Ms. Yuko Morinaga, and Ms. Eriko Koreeda for their technical assistance and data management (belonging to Digestive and Lifestyle Diseases, Department of Human and Environmental Sciences, Kagoshima University Graduate School of Medical and Dental Sciences).

## Author Contributions

**Conceptualization:** Seiichi Mawatari, Kotaro Kumagai, Kohei Oda, Kazuaki Tabu, Shuji Kanmura, Akio Ido.

**Data curation:** Seiichi Mawatari, Kotaro Kumagai, Kohei Oda, Kazuaki Tabu, Sho Ijuin, Kunio Fujisaki, Shuzo Tashima, Yukiko Inada, Hirofumi Uto, Akiko Saisyoji, Yasunari Hiramine, Masafumi Hashiguchi, Tsutomu Tamai, Takeshi Hori, Ohki Taniyama, Ai Toyodome, Haruka Sakae, Takeshi Kure, Kazuhiro Sakurai, Akihiro Moriuchi, Shuji Kanmura.

**Formal analysis:** Seiichi Mawatari.

**Funding acquisition:** Seiichi Mawatari, Akio Ido.

**Investigation:** Seiichi Mawatari, Kotaro Kumagai, Kohei Oda, Kazuaki Tabu, Sho Ijuin, Kunio Fujisaki, Shuzo Tashima, Yukiko Inada, Hirofumi Uto, Akiko Saisyoji, Yasunari Hiramine, Masafumi Hashiguchi, Tsutomu Tamai, Takeshi Hori, Ohki Taniyama, Ai Toyodome, Haruka Sakae, Takeshi Kure, Kazuhiro Sakurai, Akihiro Moriuchi, Shuji Kanmura, Akio Ido.

**Project administration:** Seiichi Mawatari.

**Resources:** Seiichi Mawatari.

**Supervision:** Shuji Kanmura, Akio Ido.

**Writing – original draft:** Seiichi Mawatari.

**Writing – review & editing:** Seiichi Mawatari, Akio Ido.

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
