## [Decision Letter · Decision Letter 0]

21 Oct 2021

PONE-D-21-28792Features of patients who developed hepatocellular carcinoma after direct-acting antiviral treatment for hepatitis C VirusPLOS ONE

Dear Dr. Mawatari,

Thank you for submitting your manuscript to PLOS ONE. After careful consideration, we feel that it has merit but does not fully meet PLOS ONE’s publication criteria as it currently stands. Therefore, we invite you to submit a revised version of the manuscript that addresses the points raised during the review process.

Please comply with all points raised by the reviewers. In particular, please address the issue of  underdiagnosis of cirrhosis as it will very likely impact accurate identification of factors associated with HCC as well as development of models for HCC risk assessment. As suggested, colinearity between HA and chirrosis/advance fibrosis should be explored and included into the dataset. Furthermore, the issues raised in respect to confounding factors in the cohort should be addressed.

We look forward to receiving your revised manuscript.

Kind regards,

Birke Bartosch

Academic Editor

PLOS ONE

Journal Requirements:

“This work was supported in part by a grant-in-aid from the Ministry of Health, Labour and Welfare of Japan (grant number: 18K15821).”

“This work was supported in part by a grant-in-aid from the Ministry of Health, Labour and Welfare of Japan (grant number: 18K15821).”

We note that you have provided additional information within the Acknowledgements Section. Please note that funding information should not appear in the Acknowledgments section or other areas of your manuscript. We will only publish funding information present in the Funding Statement section of the online submission form.

 “This work was supported in part by a grant-in-aid from the Ministry of Health, Labour and Welfare of Japan (grant number: 18K15821).”

“I have read the journal's policy and the authors of this manuscript have the following competing interests: AI received honoraria for lectures from Bristol-Myers Squibb Co., Ltd., MSD Co., Ltd., Gilead Sciences Co., Ltd., and Abbvie Inc., and received research funding from Eisai Co., Ltd., Bristol-Myers Squibb Co., Ltd, MSD Co., Ltd., and Abbvie Inc.

The other authors declare no conflicts of interest in association with the present study.”

Please include your updated Competing Interests statement in your cover letter; we will change the online submission form on your behalf

7. We note that you have included the phrase “data not shown” in your manuscript. Unfortunately, this does not meet our data sharing requirements. PLOS does not permit references to inaccessible data. We require that authors provide all relevant data within the paper, Supporting Information files, or in an acceptable, public repository. Please add a citation to support this phrase or upload the data that corresponds with these findings to a stable repository (such as Figshare or Dryad) and provide and URLs, DOIs, or accession numbers that may be used to access these data. Or, if the data are not a core part of the research being presented in your study, we ask that you remove the phrase that refers to these data.

Reviewers' comments:

Reviewer's Responses to Questions

**Comments to the Author**

1. Is the manuscript technically sound, and do the data support the conclusions?

Reviewer #1: Partly

Reviewer #2: Yes

Reviewer #3: Yes

2. Has the statistical analysis been performed appropriately and rigorously? 

Reviewer #1: Yes

Reviewer #2: Yes

Reviewer #3: Yes

3. Have the authors made all data underlying the findings in their manuscript fully available?

Reviewer #1: Yes

Reviewer #2: Yes

Reviewer #3: Yes

4. Is the manuscript presented in an intelligible fashion and written in standard English?

Reviewer #1: Yes

Reviewer #2: Yes

Reviewer #3: Yes

5. Review Comments to the Author

Reviewer #1: In the current study, Mawatari et aimed to shed light on the features of HCV-infected patients developing HCC upon DAA treatment. After studying a cohort of 1494 DAA-SVR patients, they found that 60 of them (=4%) developed HCC within 47.6 months and in these patients. The main conclusion is that AFP>5.3 ng/mL at the end of the DAA treatment (EOT) can be useful for predicting HCC development after an SVR. Eventually, the combination of the following markers is associated for predicting HCC: EOT-AFP>5.3 ng/mL, males aged >73 years, hyaluronic acid >75 ng/mL, EOT-albumin <3.9 g/dL.

The study is essentially descriptive and statistically sound. However, I have some concerns about the usefulness of these markers in predicting HCC in DAA treated patients:

1. These markers have not been evaluated in a cohort without DAA treatments (par ex. NASH, NAFLD)

2. I have doubts about the age and sex considered as risk factors in this study since higher HCC rates are encountered in older males irrespectively of HCC etiology.

3. The authors state that EOT-AFP is an important indicator for HCC development, but at the same time they state in table 5 that in 37.3% of the patients who developed HCC, the AFP (and DCP) values were within the normal limits. This is confusing. Also, DCP has not been explained and commented.

4. Other markers that could be of interest upon HCV clearance such as miRNA-122 have not been studied and discussed. Immune markers such as TGF-beta, VEGF, IL-6, etc. have not been addressed neither.

Reviewer #2: In this manuscript, Mawatari and colleagues analyzed the features of patients with chronic C Hepatitis successfully treated with DAAs who developed HCC over time.

He conducted a well-designed prospective study, with a large study population and considerable follow-up.

Although the parameters (including AFP levels) associated with HCC development after DAAs therapy have been already analyzed and described by several authors, changes in AFP levels over time after DAAs were not considered before and represent therefore a novel parameter of interest.

The study has a rationale and has been carried out with an adequate statistical approach.

The article address a an important topic, considering that HCV infection is now easily cured with IFN-free therapies, but HCC risk still remains after achieving SVR. The authors analyzed the risk factors for HCC and created a 5- factors score that could be use to estimate the cumulative HCC risk over time, categorizing the patients in low-moderate-high risk. The laboratory parameters used in this score system are routinely tested in clinics, therefore the score, if further validated, could be suggested as tool in clinical practice.

The fact that liver fibrosis was not diagnosed by biopsy or transient elastography represents a limitation, but the authors acknowledged it. A validation of this score will be important in order to understand its potential role in clinical practice in earlier identify subjects at higher risk.

Major comments:

• In the introduction, the authors explain the concept of RASs in relationship to non-response to DAAs. I don’t find this particular topic related to the main subject of the manuscript since only the patients who achieved SVR were enrolled. This paragraph should be removed.

• The authors do not mention HIV-coinfected patients, who have overall a higher risk for cancer. Were they excluded from the study population? If so, the authors should mention that in the Methods. Otherwise, they should consider this variable.

• Is the study still ongoing? It is well specified when the observation started but it’s not clear to me if the observation period is over.

• Patients with EOT-AFT<5.3 ng/ml did not show any changes in AFP levels at HCC onset. Did the author observe any peculiar characteristics (size, number of nodules, vascularization?) in HCC occurred in this subgroup of patients?

Minor comments:

Material and Methods:

-“observation” should be replaced by “observational”

- Instead of “The study population is shown in Figure 1”, I would rather write “Study population enrollment is shown…” I would suggest also to modify Figure 1 title consistently with that, specifying that it represents a flow chart of the study population enrollment.

- Titles of Table 3 and 4 are the same. Authors should be differentiate.

Reviewer #3: This is a Japanese multicentre cohort study of n=1494 post-SVR patients following DAA treatment. The study followed the cohort (median 47-months) and described the factors associated with the development of HCC (N=60). They identify: age, males, hyaluronic acid, AFP and albumin as factors associated with HCC. They then developed a risk stratification score based on these factors.

The manuscript is well structured and clear. A strength of the study is the exploration of AFP dynamics. This explores the impact of change in AFP and then relationship to DCP. This supports the data that AFP is not a great screening tool.

Comments:

•The major issue with this study is the categorisation of cirrhosis. It is clear from multiple studies that cirrhosis is a major aetiological factor in HCC development and the vast majority of HCCs occur in the context of cirrhosis. In this study only 55% of HCC was associated with cirrhosis which suggests that cirrhosis was underdiagnosed. It is likely the result of suboptimal assessment of liver fibrosis. This would impact any analysis of factors associated with HCC, especially if they are factors that are also related to fibrosis/cirrhosis. Key to this is hyaluronic acid which is a strong marker of fibrosis. The authors were not clear about their definition of cirrhosis and this needs to be stated clearly in the methods. Specifically, cut-offs for FIB4, clinical and biochemical criteria etc. It is not acceptable to state “other factors”.

•Hyaluronic acid (HA) is related to liver fibrosis. It is a part of many non-invasive algorithms of liver fibrosis. Given this, it is likely closely related to cirrhosis. Potentially it is collinear with cirrhosis. In the 2-year follow-up and 3-year follow-up models HA and cirrhosis are alternatively significant. Collinearity between HA and cirrhosis and advanced fibrosis must be explored.

•Gender and age differences in HA may also have an impact. Could this be analysed in the cohort?

•HA should also form a part of the discussion.

•Usually, to analyse de novo HCC, any HCC exclusion that developed within 3-6 months post treatment should be excluded. Could this be done to see if this changes the analysis of factors.

•A weakness is co-factors which could not be addressed including: alcohol, metabolic syndrome, and aspirin use. This would lead to confounding of results.

•This is a Japanese population with different screening protocols compared with other international guidelines. Here, even non-cirrhotic patients were surveyed and cirrhotics were evaluated at a 3-4 month interval when most international guidelines are 6-months US. This should be made clear in the discussion as it will impact generalisability.

Minor:

•P15 – The comparison is cirrhosis to no cirrhosis (not chronic hepatitis as the HCV has been cleared).

•Table 3: Aetiology should be cirrhosis (present or not)

•Table 3: In the Advanced fibrosis multivariable analysis, why are insignificant outcomes presented?

•Figure 2 should change the y-axis to small scale.

6. PLOS authors have the option to publish the peer review history of their article (what does this mean?). If published, this will include your full peer review and any attached files.

Reviewer #1: No

Reviewer #2: **Yes: **Viola Guardigni

Reviewer #3: **Yes: **mark danta

---

## [Author Response · Author response to Decision Letter 0]

20 Nov 2021

Thank you for your valuable feedback. We have revised the manuscript based on the comments.

Response to reviewer 1

1. These markers have not been evaluated in a cohort without DAA treatments (par ex. NASH, NAFLD)

Response: In the present study, we aimed to clarify the features of patients who developed HCC after an SVR with DAA treatment. Therefore, we could not evaluate the carcinogenic features in a cohort without DAA treatment (e.g., NASH, NAFLD). In addition, various factors (alcohol intake, obesity, etc.) were not examined after treatment in this cohort. We have mentioned this in the Discussion section.

2. I have doubts about the age and sex considered as risk factors in this study since higher HCC rates are encountered in older males irrespectively of HCC etiology.

Response: As you pointed out, it is well-known that older males are at increased risk for the development of HCC. We believe that these factors should always be considered in a statistical analysis. In fact, this cohort demonstrates that “older male” is an independent risk factor for the development of HCC.

3. The authors state that EOT-AFP is an important indicator for HCC development, but at the same time they state in table 5 that in 37.3% of the patients who developed HCC, the AFP (and DCP) values were within the normal limits. This is confusing. Also, DCP has not been explained and commented.

Response: AFP and DCP are tumor markers for HCC. We therefore confirmed the AFP and DCP levels at the onset of HCC to consider whether these markers were useful for carcinogenesis monitoring. More details can be found in the following section (Alpha-fetoprotein and des-γ-carboxy prothrombin levels at the onset of HCC) on page 19. What we wanted to insist in this chapter is that in patients with EOT-AFP <5.3 ng/mL, AFP did not increase, even if they developed HCC. On the other hand, 27 patients (45.8%) had DCP levels above the upper limit (40 mAU/mL) at the onset of HCC. In brief, DCP at the onset of HCC was not affected by EOT-AFP, it is considered useful to measure DCP in the clinical course. However, 22 patients (37.3%) had AFP and DCP levels within the normal limits at the onset of HCC. In particular, 57.1% of patients with EOT-AFP<5.3 ng/mL had AFP and DCP levels within the normal limits at the onset of HCC. It seems difficult to predict carcinogenesis in patients who have achieved a DAA-SVR from tumor markers alone. We discussed these details on page 22–23. 

We added the baseline data about DCP. There was no statistically significance in the des-γ-carboxy prothrombin (DCP) level; however, there were many missing data.

4. Other markers that could be of interest upon HCV clearance such as miRNA-122 have not been studied and discussed. Immune markers such as TGF-beta, VEGF, IL-6, etc. have not been addressed neither.

Response: Thank you for your suggestion. Unfortunately, we could not measure the miRNA, TGF-beta, VEGF, and IL-6 levels. We added a comment about these markers to the Discussion section. We added the following statement to the Discussion section:

HCC occurs as a result of hepatic inflammation and/or changes in the tumor microenvironment. HCV clearance can suppress fibrosis and reduces the incidence of HCC. It is assumed that microRNAs (e.g., miRNA-122) and inflammatory cytokines (e.g., transforming growth factor [TGF]-beta, vascular endothelial growth factor [VEGF], and interleukin-6) are involved in this inflammation, and microRNAs are involved in HCV clearance and HCC development. HCV clearance limits fibrosis and reduces the incidence of HCC by switching TGF-beta signaling from fibro-carcinogenesis to tumor suppression. Otherwise, the administration of DAAs induces an early increase in serum VEGF and a change in the inflammatory pattern, which coincides with HCV clearance. In this study, we have not been able to investigate the onset of HCC with microRNAs and inflammatory cytokines; however, we consider this to be an issue for future study.

Response to reviewer 2

Major comments:

• In the introduction, the authors explain the concept of RASs in relationship to non-response to DAAs. I don’t find this particular topic related to the main subject of the manuscript since only the patients who achieved SVR were enrolled. This paragraph should be removed.

Response: We have removed the concept of RASs.

• The authors do not mention HIV-coinfected patients, who have overall a higher risk for cancer. Were they excluded from the study population? If so, the authors should mention that in the Methods. Otherwise, they should consider this variable.

Response: Thank you for your suggestion. In this study, we did not mention HIV-co-infected patients. There were no patients who received therapy for HIV. A nationwide survey in Japan revealed that nearly one-fifth of HIV-positive patients are co-infected with HCV. The determination of HCV genotypes revealed that genotype 3 or 4, which is rarely seen in HCV mono-infected patients in Japan, was found in a substantial fraction of HIV-infected patients. This cohort did not include patients with HCV genotype 3 or 4. We therefore considered that there were no HIV-HCV co-infected patients in this cohort.

We added the following statement in the Discussion section.

HIV seropositivity accelerates the progression of fibrosis related to chronic hepatitis C. (Benhamou). A nationwide survey in Japan revealed that nearly one-fifth of HIV-positive patients are co-infected with HCV. The determination of HCV genotypes revealed that genotype 3 or 4, which is rarely seen in HCV mono-infected patients in Japan, was found in a substantial fraction of HIV-infected patients. In this cohort, there were no patients who received the therapy for HIV, and with HCV genotype 3 or 4. We therefore considered that there were no HIV-HCV co-infected patients in this cohort.

• Is the study still ongoing? It is well specified when the observation started but it’s not clear to me if the observation period is over.

Response: The study is still ongoing. The patients were observed until July 2021. We added this information to the Materials and Methods section. 

• Patients with EOT-AFT<5.3 ng/ml did not show any changes in AFP levels at HCC onset. Did the author observe any peculiar characteristics (size, number of nodules, vascularization?) in HCC occurred in this subgroup of patients?

Response: We have added the characteristics of HCC size, number of nodules, and vascularization in Table 6. The tumor size, number, and vascularity did not differ between the two groups to a statistically significant extent.

Minor comments:

Material and Methods:-“observation” should be replaced by “observational”

Response: We changed “observation” to “observational”.

- Instead of “The study population is shown in Figure 1”, I would rather write “Study population enrollment is shown…” I would suggest also to modify Figure 1 title consistently with that, specifying that it represents a flow chart of the study population enrollment.

Response: We changed the sentence to “Study population enrollment is shown in Figure 1”, and modified the Figure 1 title accordingly. 

- Titles of Table 3 and 4 are the same. Authors should be differentiate.

Response: We added subtitles to Tables 3, 4, and 5.

Response to reviewer 3

Comments:

•The major issue with this study is the categorisation of cirrhosis. It is clear from multiple studies that cirrhosis is a major aetiological factor in HCC development and the vast majority of HCCs occur in the context of cirrhosis. In this study only 55% of HCC was associated with cirrhosis which suggests that cirrhosis was underdiagnosed. It is likely the result of suboptimal assessment of liver fibrosis. This would impact any analysis of factors associated with HCC, especially if they are factors that are also related to fibrosis/cirrhosis. Key to this is hyaluronic acid which is a strong marker of fibrosis. The authors were not clear about their definition of cirrhosis and this needs to be stated clearly in the methods. Specifically, cut-offs for FIB4, clinical and biochemical criteria etc. It is not acceptable to state “other factors”.

Response: We stated the definition of cirrhosis as follows:

Liver cirrhosis was comprehensively judged by hepatologists at each institution according to the platelet count, imaging, fibrosis markers, transient elastography, or varices formation. Unfortunately, many facilities have not been able to investigate elastography. As you indicated, cirrhosis was likely to be underdiagnosed in this study. We added this to the limitations. Hyaluronic acid is a fibrosis marker and has been adopted in scoring systems for the diagnosis of liver cirrhosis (e.g., ELF score, Hepascore etc.); however, in this study, the parameters for calculating these scores were insufficient. We therefore added the cases with Fib-4 index ≥ 3.25, who were diagnosed with severe fibrosis, to the baseline data (Table 1).

•Hyaluronic acid (HA) is related to liver fibrosis. It is a part of many non-invasive algorithms of liver fibrosis. Given this, it is likely closely related to cirrhosis. Potentially it is collinear with cirrhosis. In the 2-year follow-up and 3-year follow-up models HA and cirrhosis are alternatively significant. Collinearity between HA and cirrhosis and advanced fibrosis must be explored.

Response: Thank you for your comments. We added the collinearity between HA and cirrhosis and advanced fibrosis (e.g., Fib-4 index ≥ 3.25). 

•Gender and age differences in HA may also have an impact. Could this be analysed in the cohort?

Response: We analyzed the association between sex and age and HA in Fig.5. HA was associated with age; however, there was no significant association between HA and sex. 

•HA should also form a part of the discussion.

Response: In the Discussion section, we discussed HA as follows:

HA is an acidic mucopolysaccharide that is widely distributed in the connective tissue of the body, and is produced in the liver by activated hepatic stellate cells. During chronic liver inflammation, there is continuous hepatic stellate cell activation and therefore increased HA synthesis. HA is related to liver fibrosis. It is a part of many non-invasive algorithms that are used to asses liver fibrosis (e.g., ELF score and Hepascore). In this study, the HA levels of patients with liver cirrhosis were higher than those of non-cirrhotic patients and showed a positive correlation with the Fib-4 index (Fig 5). A previous report showed that the initiation of liver cancer requires the inhibition of p53 by CD44 (a receptor of hyaluronic acid)-enhanced growth factor signaling. HA can be a predictive marker for the development of HCC after DAA-SVR.

•Usually, to analyse de novo HCC, any HCC exclusion that developed within 3-6 months post treatment should be excluded. Could this be done to see if this changes the analysis of factors.

Response: We added the model of HCC exclusion that developed within 6 months and 1 year in Table 4. With the exception of albumin, the factors associated with carcinogenesis were similar.

•A weakness is co-factors which could not be addressed including: alcohol, metabolic syndrome, and aspirin use. This would lead to confounding of results.

Response: As you pointed out, we could not address co-factors, including alcohol, metabolic syndrome, and aspirin use. We have described this as a limitation of this study. 

•This is a Japanese population with different screening protocols compared with other international guidelines. Here, even non-cirrhotic patients were surveyed and cirrhotics were evaluated at a 3-4 month interval when most international guidelines are 6-months US. This should be made clear in the discussion as it will impact generalisability.

Response: Thank you for your comments. In the Discussion section, we added the following text:

This study performed HCC screening based on the Japanese guidelines. The American Association for the Study of Liver Diseases (AASLD), The European Association for the Study of the Liver (EASL), and The Asian Pacific Association for the Study of the Liver (APASL) guidelines recommend ultrasound screening every 6 months. In particular, the AASLD guideline shows that the risk of HCC is significantly lower in patients without cirrhosis in comparison to those with cirrhosis, and surveillance is not recommended for these patients. However, in this study, out of 60 patients who developed HCC, 27 patients (45%) were non-cirrhotic and 15 (25%) had a Fib-4 index of <3.25 (Table 1); thus, HCC surveillance after DAA-SVR should be considered important, even for non-cirrhotic patients. The EASL Clinical Practice Guidelines show that non-cirrhotic F3 patients, regardless of etiology, may be considered for surveillance based on an individual risk assessment, and in patients who were treated viral chronic hepatitis, there was no evidence for a timing or stiffness threshold to stop surveillance in patients who were included in surveillance programs. In this study, a few patients in the low and moderate-risk groups developed HCC. In the future, it will be necessary to consider the cases in which surveillance can be stopped after DAA-SVR. 

Minor:

•P15 – The comparison is cirrhosis to no cirrhosis (not chronic hepatitis as the HCV has been cleared).

Response: We changed cirrhosis to no cirrhosis

•Table 3: Aetiology should be cirrhosis (present or not)

Response: We changed cirrhosis to cirrhosis present in Tables 3, 4 and 5.

•Table 3: In the Advanced fibrosis multivariable analysis, why are insignificant outcomes presented?

Response: Factors associated with HCC development were determined using a Cox proportional hazards analysis with forward selection using p<0.10 as a cutoff value for inclusion in the model. The above is described in the Statistical analyses section. 

•Figure 2 should change the y-axis to small scale.

Response: We changed the y-axis to a smaller scale in Figure 2.

---

## [Decision Letter · Decision Letter 1]

21 Dec 2021

Features of patients who developed hepatocellular carcinoma after direct-acting antiviral treatment for hepatitis C Virus

PONE-D-21-28792R1

Dear Dr. Mawatari,

We’re pleased to inform you that your manuscript has been judged scientifically suitable for publication and will be formally accepted for publication once it meets all outstanding technical requirements.

Kind regards,

Birke Bartosch

Academic Editor

PLOS ONE

Additional Editor Comments (optional):

Reviewers' comments:

Reviewer's Responses to Questions

**Comments to the Author**

1. If the authors have adequately addressed your comments raised in a previous round of review and you feel that this manuscript is now acceptable for publication, you may indicate that here to bypass the “Comments to the Author” section, enter your conflict of interest statement in the “Confidential to Editor” section, and submit your "Accept" recommendation.

Reviewer #1: All comments have been addressed

Reviewer #2: All comments have been addressed

Reviewer #3: All comments have been addressed

2. Is the manuscript technically sound, and do the data support the conclusions?

Reviewer #1: Yes

Reviewer #2: Yes

Reviewer #3: Yes

3. Has the statistical analysis been performed appropriately and rigorously? 

Reviewer #1: I Don't Know

Reviewer #2: Yes

Reviewer #3: Yes

4. Have the authors made all data underlying the findings in their manuscript fully available?

Reviewer #1: Yes

Reviewer #2: Yes

Reviewer #3: Yes

5. Is the manuscript presented in an intelligible fashion and written in standard English?

Reviewer #1: Yes

Reviewer #2: Yes

Reviewer #3: Yes

6. Review Comments to the Author

Reviewer #1: The authors have improved the manuscript according to the majority of the referees' comments and their article is now better structured and scientifically sound. The authors now clearly show the limitation of their study and the results/conclusions are therefore better interpreted.

Reviewer #2: The authors have addressed each of my previuos comments exhaustively. I have no further comment to add concerning the contents. I think that the manuscript could be accepted for publication on PLOS ONE.

Reviewer #3: The authors have addressed the issues that were raised as best they could. There are still issues around confounding and classification of cirrhosis which could not be addressed in the study design.

7. PLOS authors have the option to publish the peer review history of their article (what does this mean?). If published, this will include your full peer review and any attached files.

Reviewer #1: No

Reviewer #2: **Yes: **Viola Guardigni

Reviewer #3: **Yes: **A/Prof Mark Danta

---

## [Editor Report · Acceptance letter]

26 Dec 2021

PONE-D-21-28792R1 

Features of patients who developed hepatocellular carcinoma after direct-acting antiviral treatment for hepatitis C Virus 

Dear Dr. Mawatari:

I'm pleased to inform you that your manuscript has been deemed suitable for publication in PLOS ONE. Congratulations! Your manuscript is now with our production department. 

Kind regards, 

on behalf of

Dr. Birke Bartosch 

Academic Editor

PLOS ONE